# A systematic review of non-clinician trauma-based interventions for school-age youth

**Flo Avery**[1]*, **Natasha Kennedy**[1], **Michaela James**[1], **Hope Jones**[1], **Rebekah Amos**[2], **Mark Bellis**[3], **Karen Hughes**[2,4], **Sinead Brophy**[1]

**1** National Centre for Population Health and Wellbeing, School of Medicine, Swansea University, Swansea, United Kingdom, **2** School of Human Sciences, Bangor University, Wrexham, United Kingdom, **3** Faculty of Health, Liverpool John Moores University, Liverpool, United Kingdom, **4** Policy and International Health Directorate, World Health Organization Collaborating Centre on Investment for Health and Well-Being, Public Health Wales, Wrexham, United Kingdom

\* 2226345@swansea.ac.uk

**Data Availability Statement:** This work is based on publicly available papers and so is in the public domain. Further details of the search strategy and quality assessment have been made publicly

## Abstract

Exposure to adverse childhood experiences (ACEs) is recognised globally as a risk factor for health problems in later life. Awareness of ACEs and associated trauma is increasing within schools and educational settings, as well as the demand for supportive services to address needs. However, there is a lack of clear evidence for effective interventions which can be delivered by non-clinicians (e.g., the school staff themselves). Thus, we undertook a systematic review to answer the question: What evidence exists for the efficacy of non-clinician delivered trauma-based interventions for improving mental health in school-age youth (4–18 years) who have experienced ACEs? The protocol for the review is registered in the PROSPERO International Prospective Register of Systematic Reviews (ID: CRD42023417286). We conducted a search across five electronic databases for studies published between January 2013 and April 2023 that reported on interventions suitable for non-clinician delivery, were published in English in the last 10 years, and involved participants aged 4–18 years (school-age) that had exposure to ACEs. Of the 4097 studies identified through the search, 326 were retrieved for full text screening, and 25 were included in the final review. Data were extracted from included articles for analysis and selected studies were quality assessed using validated assessment tools. Data were analysed through narrative synthesis. There was considerable heterogeneity in study design, outcome measures, and the interventions being studied. Interventions included CBT, mindfulness and art-based programs. A key finding was that there is a lack of high-quality research evidence to inform non-clinician delivered trauma-informed interventions. Many included studies were weak quality due to convenience sampling of participants and potential bias. Cognitive Behavioural Therapy (CBT)-based approaches are tentatively suggested as a suitable target for future rigorous evaluations of interventions addressing ACE-related trauma recovery and mental health improvement in school-age youth.

available and can be found at https://datadryad.org/stash/dataset/doi:10.5061/dryad.x3ffbg7sc.

**Funding:** This research was supported by the National Centre for Population Health and Wellbeing Research. Funding was provided by Public Health Wales and the Economic and Social Research Council (ESRC) through their support of a PhD studentship. The funders had no role in study design, data collection and analysis, decision to publish, or preparation of the manuscript.

**Competing interests:** The authors have declared that no competing interests exist.

## Introduction

Adverse Childhood Experiences (ACEs) are potentially traumatic experiences, such as child abuse or neglect, or household dysfunction (such as drug and alcohol abuse or domestic violence within the family home), occurring before the age of 18 [1]. Young people exposed to ACEs face an elevated risk of unhealthy behaviours and long term physical and mental health problems [2]. This correlation is particularly pronounced among those who have experienced multiple ACE types [3]. ACEs are common with around half of adults in general populations having suffered at least one ACE [4]. This is a significant proportion of the population, suggesting a universal approach to offering support for ACEs may be appropriate.

Promoting awareness of ACEs and the potentially traumatic effects of these experiences is increasingly recommended both in healthcare [5] and educational [6] settings. Schools and educational settings are recognised as crucial settings for providing mental health support to young people [7]. Relatedly, there have been calls for policies to support students experiencing trauma [8]. This is particularly important for schools which serve low-income areas, where exposure to ACEs among student populations is likely to be higher, although ACEs are experienced by children across all socio-economic groups [9]. Regardless of whether a disclosure of ACEs has been made, or whether there is a clinical diagnosis of PTSD, mental health difficulties arising from ACEs are an important public health challenge for school age youth.

ACEs are well described as potential sources of trauma for children. Numerous reviews have examined the evidence for different systemic approaches and therapeutic interventions for supporting youth who have had potentially traumatic experiences [10–12]. However, many of the interventions in these reviews are clinical in nature, delivered by psychologists, therapists, or medical and allied health professionals. There is a notable gap in evidence for interventions suitable for non-clinician delivery [13]. This is a barrier for many schools, which have limited access to clinicians such as counsellors and psychologists, and do not have mental health professionals on staff. Franklin et al. [14] have suggested that interventions delivered by teachers and non-clinicians can be beneficial for youth mental health, they highlight the lack of empirical evidence regarding their effectiveness. Nonetheless, teachers and teaching assistants are increasingly tasked with providing mental health support to young people [15, 16]. Ensuring that schools have access to evidence-based interventions deliverable by staff for supporting those who have experienced ACEs is important, but presently no summary of this evidence exists.

The review question was: what evidence exists for the efficacy of non-clinician delivered trauma-based interventions for improving mental health in school-age youth (4–18 years) who have experienced ACEs? The objective was to synthesise evidence from a systematic literature review process for interventions which are appropriate for professionals such as teachers and teaching assistants.

## Methods

This systematic review was conducted in accordance with the Preferred Reporting Items for Systematic Reviews and Meta-Analysis (PRISMA) guidelines [17]. The protocol for the review was registered in the PROSPERO International Prospective Register of Systematic Reviews in June 2023 (ID: CRD42023417286). No changes have been made to the registered protocol.

### Review design

We conducted a systematic review of quantitative and qualitative evidence to evaluate the effectiveness of trauma-based interventions to support young people with ACEs which are

suitable for non-clinician delivery. The study findings were summarised through a narrative synthesis.

## Search strategy

Five electronic databases–Web of Science, Embase, Science Direct, Applied Social Sciences Index (ASSIA) and EBSCO (including CINAHL Plus with Full Text, MEDLINE, APA PsycArticles, APA PsycInfo, Teacher Reference Center, and Education Research Complete)–were searched to identify relevant studies published in English. The literature search was conducted in April 2023. Search terms included: Trauma* OR "Post-Traumatic Stress" OR PTSD and Intervention* OR Treatment* and children OR youth OR young OR adolescen* and education OR school OR teach* OR play. Detailed search terms can be viewed in S1 File.

## Eligibility

**Inclusion criteria.** Articles were included if they met the following criteria: (1) published in English in the last 10 years (e.g. since 1st January 2013); (2) reported on a supportive or therapeutic intervention with applicability to trauma recovery; (3) intervention was suitable for non-clinician delivery e.g. teacher, teaching assistant or similar; (4) participants were aged 4–18 years (school-age) with any experience of or exposure to ACEs; (5) intervention took place in a school, educational setting, community setting, residential or care settings in any country; and (6) included a validated self-reported mental health outcome relating to trauma or adversity such as post-traumatic stress disorder (PTSD) symptoms as an outcome measure. Both randomised and non-randomised studies were included. Language of publication was restricted to English owing to constraints of the research team's capacity. Publication date restrictions were imposed to ensure the relevance of the included studies.

**Exclusion criteria.** Articles were excluded if they met the following criteria: 1) All participants were over 18 or under 4 years; 2) interventions were assessed indirectly through a teacher or professional assessment of participants; 3) interventions were based on disasters such as earthquakes, floods or experiences of war; 4) interventions were delivered by clinicians; 5) interventions were behavioural rather than supportive, such as violence prevention or sex education programmes; 6) the research study had implemented a case study design; and 7) the research had been conducted in a hospital or in-patient healthcare setting.

## Study selection

The search yielded 7147 results, which were imported into Covidence for title and abstract screening. After removing 3050 duplicates, the remaining titles and abstracts (4097) were screened against the selection criteria by the primary author, FA, and removed if they did not meet the selection criteria. If it was unclear from the title and abstract whether all the selection criteria were met, the article was allocated to full-text screening. Additionally, all titles and abstracts underwent independent screening by a second reviewer. Reviewers achieved a 96% agreement rate. Any disagreements were resolved through discussions among authors (FA, SB, NK, RA, HJ, MJ), resulting in a majority consensus reached by agreement between at least three authors. Consequently, 3770 studies were excluded as irrelevant. Title and abstract screening was completed on 23 June 2023. FA retrieved and independently screened all full-text articles (326), which were also independently screened by a second reviewer. Reviewers reached an 88% agreement rate, resolving disagreements through discussion amongst authors (FA, SB, NK, HJ, MJ), with a majority vote from at least three authors. A number of articles appeared to meet the inclusion criteria, but full text screening revealed that the interventions were delivered by a non-clinician and so these were excluded. Fig 1 shows details of the

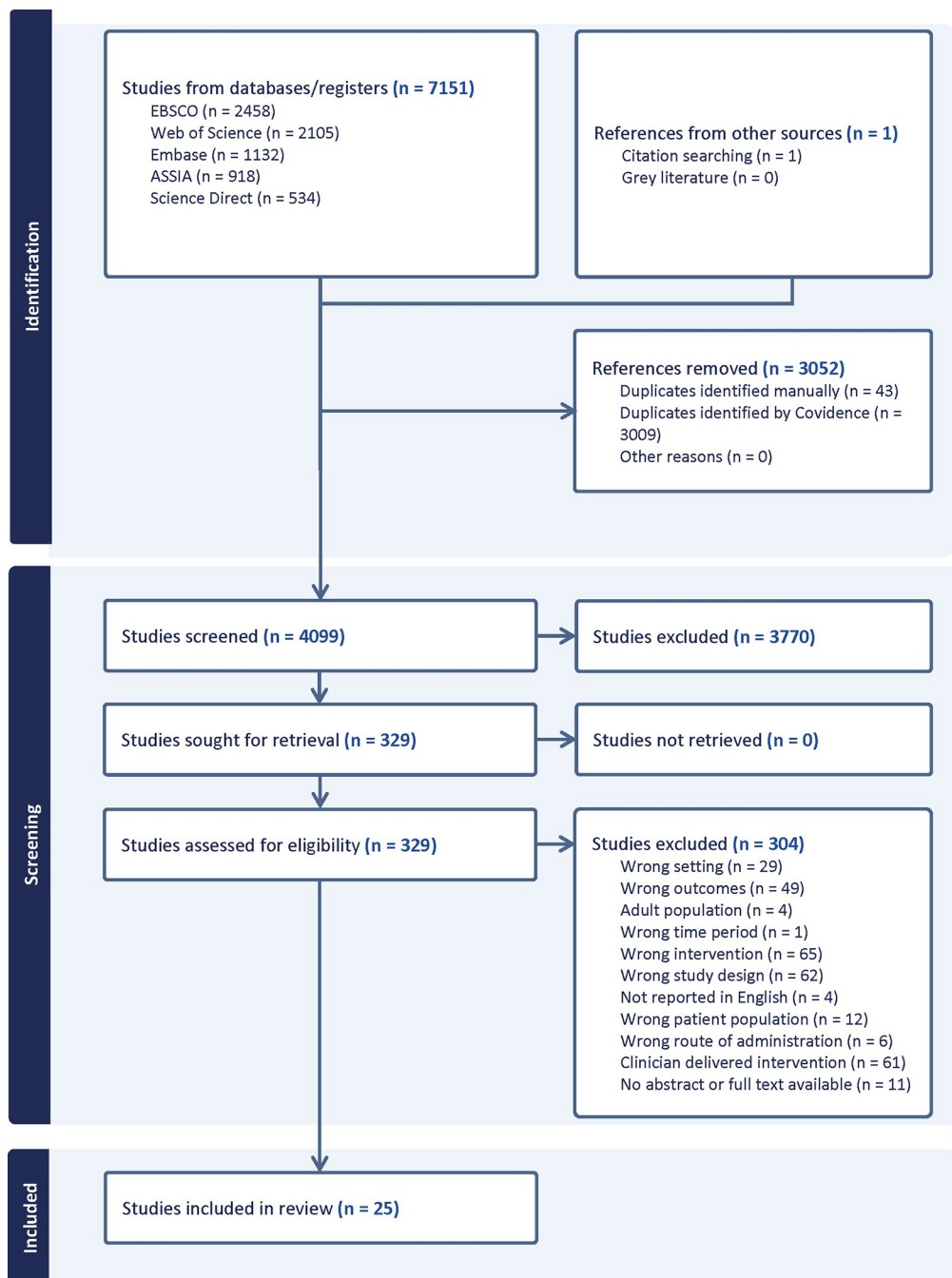

**Fig 1. PRISMA diagram showing number of studies identified, screened, and included in final review.**

selection process, including reasons for excluding articles at different stages. The full text screening process was completed on 16 July 2023. Twenty-four articles met the inclusion criteria. Data extraction from these articles was undertaken by one reviewer (FA) and cross-checked by a second reviewer (RA or MJ). Information from all articles was extracted into a data extraction table in Excel so that included studies could be compared and analysed. Data extracted included demographic data regarding the papers such as country of study, target

population and year of publication, and descriptive information about the intervention such as type of intervention, length of follow-up, measures of effect and barriers to implementation. Data extraction headings can be found in S2 File. The outcome measures differed across studies and so a descriptive extraction of the data in each study was recorded. Analysis was carried out by assessing the quality of each study using validated tools and synthesising the findings of the selected studies. Furthermore, reference lists and citations from these twenty-four studies were manually searched by FA, resulting in the identification of one additional paper. Twenty-five studies were therefore included.

## Quality assessment

Effective Public Health Practice Project (EPHPP)'s Quality Assessment Tool for Quantitative Studies, which has been validated for use in public health research [18], was used to assess the quality of quantitative research including randomised control trials (RCTs), non-randomised experimental studies, and cohort design studies. This tool assesses potential selection bias, the appropriateness of study design, confounding, blinding, data collection tools, participant dropouts, intervention integrity, and statistical analyses. Quality ratings are scored on a scale of 1–3, where 1 indicates strong quality, 2 indicates moderate quality, and 3 indicates weak quality. Overall ratings are classified as a) weak (when 2 or more components are rated as weak), b) moderate (when only one component rated as weak), or c) strong.

Qualitative studies were assessed using the Critical Appraisal Skills Programme (CASP) Qualitative Studies Checklist [19]. This checklist assesses three broad areas, such as a) the validity of the study, b) the study findings, and c) the utility of the research locally. Mixed methods research was assessed using both checklists. No studies were excluded based on the results of the quality assessment. Quality assessment was conducted by one reviewer (FA) and checked by a second reviewer (RA and MJ). The quality assessment scores of included studies can be found in S1 Table. Disagreements were resolved through discussion amongst authors.

## Data synthesis

Heterogeneity in study designs and outcome measures precluded the ability to conduct a meta-analysis. Analysis was therefore conducted through a narrative synthesis of the literature [20]. Each study was summarised in a descriptive paragraph as a starting point. Common characteristics of interventions were then identified for the purpose of grouping studies together and structuring the analysis. The relationship between the characteristics of an intervention and the study's findings, and between the characteristics of different studies were explored to identify which intervention features were likely to explain significant impact for participants.

## Results

### Overview of studies

Twenty-five studies were included in this review (Fig 1). Characteristics of the included studies are summarised in Table 1. Six studies were randomised control trials [21–26], fifteen used a cohort design (pre-test and post-test) [27–41], three used qualitative methods [42–44] and one used mixed methods (cohort design and qualitative methods) [45]. The mean sample size was 119 participants, ranging from 15 participants [39] to 565 participants [24]. Of the included studies, 14 (56%) had a sample size above 50 [21, 23, 24, 26, 27, 29, 30, 32, 34, 36–38, 41, 42]. As anticipated, there was methodological diversity in terms of study design, population (e.g. refugees, LGBTQ, youth offenders), geographic location (Middle East, USA, Europe, Asia), and measures used. Therefore, meta-analysis was not conducted.

**Table 1. Study characteristics.**

| First author (year) | Country | Purpose | Research design | Sample | Implementation | Measure for ACEs or trauma | Outcome measures (trauma recovery and mental health) | What was the outcome? |
|---|---|---|---|---|---|---|---|---|
| Akhtar (2021) | Jordan | Evaluate the effectiveness of Early Adolescent Skills for Emotions (EASE) to improve psychological distress. | Pilot randomised controlled trial | 59 Syrian children aged 10–14 years who reported psychological distress. | Seven 90 minute sessions | PTSD symptoms measured (CRIES-13) | Paediatric Symptom Checklist (PSC-35); Children's Revised Impact of Events Scale (CRIES-13); Warwick Edinburgh Mental Wellbeing Scale (WEMWBS); Psychological Sense of School Membership (PSSM) | No significant changes (underpowered feasibility trial) |
| Barnett (2020) | United States | Evaluate the impact of culture camps for Alaskan Native youth on mental health and resilience. | Cohort design (pilot pre-test post-test design) | 69 Alaska Native youth aged 13 to 18 years | Five all day sessions, daily for five days | Not measured. Population were high ACEs (native youth) | I-PANAS-SF (International Positive and Negative Affect Schedule, Short Form); Self Esteem-Interpersonal Needs: Belongingness | Significant improvement pre and post on the I-PANAS-SF (p < .001) |
| Barron (2021) | Brazil | Evaluate impact of Teaching Recovery Techniques (TRT), a CBT-based group, on PTSD. | Randomised controlled trial | 30 8–14 years engaged with youth work NGO | Five 90 minute sessions, weekly for five weeks. | PTSD symptoms measured (CRIES-13) | Children's Revised Impact of Events Scale (CRIES-13); The Moods and Feelings Questionnaire (MFQ); Posttraumatic Growth Inventory (PTGI) | Significant improvements on PTSD and Depression compared with control group (p < .01) |
| Bryant (2022) | Jordan | Evaluate the effectiveness of a group-based intervention (EASE) to improve young adolescents' mental health. | Randomised controlled trial | 471 adolescent Syrian refugees aged 10–14 years | Seven 90 minute sessions, weekly (plus three caregiver sessions) | Caregivers completed a 26-item traumatic events checklist (not validated) | Paediatric Symptom Checklist (PSC-35); Patient Health Questionnaire, adolescent version (PHQ-A); Children's Revised Impact of Events Scale (CRIES-13); Warwick Edinburgh Mental Wellbeing Scale (WEMWBS); Psychological Sense of School Membership (PSSM) | Significant reduction on the PSC-internalising scale compared to control group (p < .01) |

(*Continued*)

**Table 1.** (Continued)

| First author (year) | Country | Purpose | Research design | Sample | Implementation | Measure for ACEs or trauma | Outcome measures (trauma recovery and mental health) | What was the outcome? |
|---|---|---|---|---|---|---|---|---|
| Davis (2022) | United States | Measure the impact of an online yoga intervention on physical and mental health outcomes in adolescents. | Non-randomised experimental study | 45 high school students in rural Montana aged 15–18 years | Twelve 45 minute sessions, twice weekly for six weeks. | The Center for Youth Wellness ACE-Q Teen self-report | The Generalized Anxiety Disorder Scale (GAD-7); The Patient Health Questionnaire for Depressive Symptomology for Adolescents (PHQ-A); Connor-Davidson Resilience Scale (CD-RISC) | No significant improvements on relevant outcome measures |
| Day (2015) | United States | Assess the impact of implementing the Heart of Teaching and Learning (HTL) curriculum, an intervention designed to increase trauma-informed practices in education settings, on trauma symptomology. | Non-randomised experimental study | 70 female court-involved students in a residential treatment facility aged 11–18 years | Not specified—responsive rather than scheduled | PTSD symptoms measured: Child Report of Post-traumatic Symptoms (CROPS) | The Student Needs Survey (SNS); Child Report of Post-traumatic Symptoms (CROPS); Rosenberg Self-Esteem Scale (RSE) | Significant improvement in pre and post test scores for PTSD symptoms. Unexpected finding: unmet need in the domain of survival increased |
| Dumornay (2022) | United States | Assess the impact of a trauma-informed CBT-based skills curriculum on trauma and emotion regulation, as well as educational, employment, parenting, and life skills opportunities. | Non-randomised experimental study | 344 justice involved men aged 17–24 years | Not specified—responsive rather than scheduled | Risk assessment tool (not validated) | Difficulties in Emotion Regulation Scale (DERS) | No significant improvements on relevant outcome measures |
| El-Khani (2018) | Turkey | Assess the potential benefits of TRT + Parenting for families and whether it could reduce children's trauma related stress and mental health difficulties, and test feasibility for a RCT. | Non-randomised experimental study | 16 refugees aged 8–18 years and their families | Five 2 hour sessions, weekly for five weeks (plus two caregiver sessions). | PTSD symptoms measured (CRIES-13) | CRIES-13; Depression Self-Rating Scale for Children (DSRS); The Screen for Childhood Anxiety Related Disorders (SCARED); Strength and Difficulties Questionnaire (SDQ); | Significant improvement on the Intrusion sub-score of the CRIES, indicating reduced PTS ($p < .05$) |
| El-Khani (2021) | Lebanon | Test if children receiving TRT + parenting show improved child and caregiver mental health. | Randomised controlled trial | 565 refugees aged 9–12 years and their families | Five 2 hour sessions, weekly for five weeks (plus two caregiver sessions). | PTSD symptoms measured (CRIES-13) | CRIES-13; Depression Self-Rating Scale for Children (DSRS); The Screen for Childhood Anxiety Related Disorders (SCARED); Strength and Difficulties Questionnaire SDQ; | Significant improvement on the Intrusion sub-score of the CRIES, greatest reduction for those in the parent condition. Depression and anxiety also significantly reduced for both conditions compared to control ($p < .01$) |

*(Continued)*

**Table 1.** (Continued)

| First author (year) | Country | Purpose | Research design | Sample | Implementation | Measure for ACEs or trauma | Outcome measures (trauma recovery and mental health) | What was the outcome? |
|---|---|---|---|---|---|---|---|---|
| Elswick (2022) | United States | Evaluate the effectiveness of an culturally responsive adaptation of 10-week CBITS framework to include drumming and mentoring, on healing from trauma. | Non-randomised experimental study | 88 African refugees aged 12–18 years and their families | Twelve sessions, weekly for twelve weeks. | PTSD symptoms measured (CPSS) | Child PTSD Symptom Scale (CPSS) scale; Subjective Units of Distress Scales (SUDS); Child Intervention Rating Profile (CIRP) | Improvement in PTSD symptoms. P value not reported |
| Eruyar (2020) | Turkey | Evaluate the impact of an attachment-based intervention (Theraplay) for children who have experienced traumatic experiences. | Non-randomised experimental study | 30 Syrian refugees aged 8–14 years and their families | Eight 45 minute sessions, weekly for eight weeks. | PTSD symptoms measured (CRIES-8) | Children's Revised Impact of Events Scale (CRIES-8), Security Scale (SS) | Significant improvement in PTSD scores (p < .05) |
| Goldbach (2021) | United States | Assess the impact of Proud and Empowered (10-session small group intervention delivered in school) on mental health symptoms. | Randomised controlled trial | 44 SGMA (sexual and gender minority) high school students aged 13–18 years | Ten 45 minute sessions, weekly for ten weeks. | PTSD symptoms measured (PCL-5) | Sexual Minority Adolescent Stress Inventory (SMASI); Beck Anxiety Inventory; Beck Depression Inventory II; PTSD Checklist for DSM-5; Columbia- Suicide Severity Rating Scale | Significant improvements in minority stress, depression and suicidality measures (p < .05) |
| Greenbaum (2017) | United States | Evaluate the impact of a trauma-informed writing-based intervention for youth in detention on mental health outcomes | Non-randomised experimental study | 53 girls living in short term detention facilities aged 12–17 years | Twelve 90 minute sessions, twice weekly for six weeks. | Not measured. Population were high ACEs (incarcerated youth) | Brief Resilience Scale (BRS); Positive and Negative Affect Sched- ule–Short Form (PANAS-SF); Rosenberg Self-Esteem Scale (RSES); State Shame and Guilt Scale-Revised (SSGS-R) | Significant increase in resilience (p < .01) |
| Harden (2015) | United States | Evaluate the impact of a youth violence prevention and intervention program on trauma-exposed youth. | Qualitative research | 44 young people who had expressed concern about community violence aged 14–18 years | Thrice weekly for nine months | Partially measured— survey included items on parental separation, illness, and bereavement | Qualitative themes | Personal/ Collective Empowerment, and Post Traumatic Growth |
| Ito (2021) | Japan | Explore the effects of a short-term group mindfulness-based intervention on the mental health of adolescents who have experienced trauma. | Non-randomised experimental study | 49 Japanese adolescents aged 16.2–19 years* | One 60 minute session. | PTSD symptoms measured (IER-S) | Mindful Attention Awareness Scale (MAAS); Cognitive Fusion Questionnaire (CFQ); Kessler-6 (K6) (psychological distress); IER-S (PTS) | Significant improvements pre and post for depression, anxiety and hyperarousal posttraumatic stress (p < .05) |

(*Continued*)

**Table 1.** (Continued)

| First author (year) | Country | Purpose | Research design | Sample | Implementation | Measure for ACEs or trauma | Outcome measures (trauma recovery and mental health) | What was the outcome? |
|---|---|---|---|---|---|---|---|---|
| Jaycox (2019) | United States | Examine improvements in PTSD symptoms, anxiety, depressive symptoms, and behavioural problems following completion of LIFT (an online self-paced curriculum which includes a trauma track). | Non-randomised experimental study | 51 Hispanic and Black students aged 11.5–18.9 years* | Seven weekly sessions of variable length. | PTSD symptoms measured (CPSS) | Children's Coping Strategies Checklist; Self-Efficacy Questionnaire for Children; Child Post-Trauma Attitudes Scale; Strengths and Difficulties Questionnaire; Major Depression and Generalized Anxiety subscales of the Revised Children's Anxiety and Depression Scale; Child PTSD Symptom Scale (CPSS) | Significant improvements pre and post for depression, anxiety and PTSD symptoms ($p < .001$) |
| Li (2023) | China | Evaluate the effectiveness of trauma-focused cognitive behavioural therapy (TF-CBT) in a group format delivered by lay counsellors to children with trauma-related symptoms. | Randomised controlled trial | 234 young people with PTSD symptoms aged 9–12 years | Ten to twelve 50 minute sessions, for nine weeks. | PTSD symptoms measured (PCL-5) | UCLA PTSD Reaction Index for DSM-5 (PTSD-RI-5); PTSD Checklist-5 (PCL-5); Children Depression Inventory-Short (CDI-S); Screen for Child Anxiety Related Emotional Disorders (SCARED) | Significantly reduced PTSD, depression and anxiety immediately post-intervention ($p < .001$). No change at 3-month follow-up |
| Martin (2017) | Australia | Examine the impact of a music and group discussion intervention (Holyoake's DRUMBEAT programme) on disadvantaged adolescents' mental wellbeing, psychological distress, and post-traumatic stress symptoms. | Non-randomised experimental study | 62 adolescents aged 12.4–15.2 years* | Ten sessions, weekly for ten weeks. | PTSD symptoms measured (PCL-C) | Warwick Edinburgh Mental Wellbeing Scale (WEMWBS); Kessler 5 (K5), (3) Abbreviated Post-traumatic stress disorder (PTSD) Checklist–Civilian version (A PCL-C); Adapted Self-Reported Delinquency Scale (ASRDS) | No significant change for whole cohort. Significant improvements for boys pre and post in mental wellbeing ($p < .05$) |
| McMahon (2020) | United States | Investigate the impact of a trauma-responsive restorative justice program for youth involved with the juvenile justice system based on non-violent communication, conflict resolution, self-regulation. | Qualitative research | 51 youth of colour aged 11–18 years from economically disadvantaged neighbourhood | Twenty eight sessions, twice weekly for fourteen weeks. | Not measured. Population were high ACEs (low income neighbourhood) | Qualitative themes | Self-efficacy, Connection & conflict resolution, Empathy |

(*Continued*)

**Table 1.** (Continued)

| First author (year) | Country | Purpose | Research design | Sample | Implementation | Measure for ACEs or trauma | Outcome measures (trauma recovery and mental health) | What was the outcome? |
|---|---|---|---|---|---|---|---|---|
| Murray (2013) | Zambia | Evaluate the impact of lay-counsellor delivered Trauma Focused-Cognitive Behavioural Therapy (TF-CBT) to address trauma and stress-related symptoms in orphans and vulnerable children. | Non-randomised experimental study | 343 children from families affected by HIV AIDS, aged 5–18 years | 1–2 hour sessions, weekly for eight to twenty-three weeks. | PTSD symptoms measured (PCL-5) 51% of participants had over 4 | The Post-Traumatic Stress Disorder-Reaction Index (PTSD-RI); The SHAME Measure | Significant reduction pre and post in trauma symptoms and shame symptoms (p < .01) |
| Özden Bademci (2015) | Turkey | Assess impact of youth project (workshops and mentoring from volunteers, delivered on university campus) on participants' quality of life. | Qualitative research | 30 Istanbul street boys aged 14–17 years | ∼ 2 hour sessions, for two years. | Not measured. Population were high ACEs (homeless youth) | Qualitative themes | Trusted relationships developed, increased capacity for emotional regulation |
| Sarkadi (2018) | Sweden | Evaluate TRT in a community setting and describe the program's effects on PTSD and depression, and explore participants' experiences. | Non-randomised experimental study | 46 refugee minors aged 13–18 years | Five 90 = 120 minute sessions, weekly for six weeks (plus two caregiver sessions). | PTSD symptoms measured (CRIES-8) | Children's Revised Impact of Event Scale (CRIES-8); Montgomery–Åsberg Depression Rating Scale Self-report (MADRS-S); | Significant decrease in PTSD and depression symptoms (for those who had suicide ideation) (p < .001) |
| Schuurmans (2020) | Nether-lands | Evaluate the impact of three online game-based meditation interventions on post-traumatic symptoms in a sample of traumatized youth in residential care. | Randomised controlled trial | 15 youth in residential care with PTSD, aged 10–18 years | Twelve 15 minute sessions, twice weekly for six weeks. | PTSD symptoms measured (CRIES-13) | Physiological measures; CRIES-13; Depression Anxiety Stress Scales (DASS-21); Reactive Proactive Questionnaire (RPQ) | The game Muse showed improvements pre and post in PTS, stress and anxiety (p value not reported), no change for other games |
| Sitzer (2015) | United States | Impact of a fourteen-week multi-modal art therapy-based Wellness curriculum on preventing maladaptive responses to situational stress and trauma with at-risk youth. | Non-randomised experimental study | 40 elementary school pupils aged 9 to 12 years, referred by teachers for emotional concerns | Fourteen 1 hour sessions, weekly for fourteen weeks. | Not measured. Population were high ACEs ('many [participants] had a history of trauma.') | Wellness Inventory | Significant increases pre and post in resilience, social and emotional functioning (p < .05) |
| Taku (2017) | Japan | Evaluate the effect of a psychoeducational intervention about posttraumatic growth (PTG). | Randomised controlled trial | 67 female nursing program students, aged 14.3–19.3 years* | One 20–25 minute session. | Stressful life events list, including bereavement, abuse, assault (and others) | 21-item PTG Inventory (PTGI) | Significant increases pre and post in post-traumatic growth (p < .001) |

*Age range not stated–range given is two standard deviations from the mean.

The studies in this review either recruited the study population based on likely exposure to trauma, measured exposure to traumatic events, or measured PTSD symptoms (or a mix of these). Eight studies directly measured exposure to traumatic events [23, 26, 28, 30, 32, 38, 41, 43]. Eight studies directly measured PTSD symptoms and used this as selection criteria [23, 24, 26, 31, 32, 35, 39, 46]. Eight further studies directly measured PTSD symptoms [22, 25, 29, 33, 36–38, 45]. Five studies did not measure either of these but recruited the study population based on likely exposure to trauma e.g. homeless youth or incarcerated youth [27, 34, 40, 42, 44].

Ten studies assessed interventions which were delivered in the school setting [24–26, 29, 31, 32, 35, 37, 40, 41]. Eight studies looked at interventions delivered in a community setting (such as a church or charity site) [21–23, 27, 30, 33, 38, 42] two were interventions delivered on university campuses [43, 44], and one was an intervention delivered in a youth residential care setting [34]. One study evaluated the intervention across different settings, including a school and residential setting [45]. All these interventions were delivered in person. Three studies observed interventions delivered online, including an online yoga intervention accessed at home [28], a game-based meditation accessed in a residential care setting (where participants lived) [39], and a self-guided online programme which participants accessed during school hours [36].

**Quality of included studies.** The quality assessment scores of included studies can be found in S1 Table. The methodological quality of included studies was weak overall. Using the EPHPP tool for quantitative studies, two studies were categorised as strong, seven studies as moderate, and the rest as weak. Most studies rated as weak relied on convenience samples of participants and did not adequately describe methodological information, such as potential confounding and blinding. Overall, qualitative studies were not considered strong, primarily because they lacked consistent discussions on how the relationship between researcher and participants was managed, including consideration of power dynamics and ethical issues.

The nine studies rated moderate or strong on quality included a range of interventions. Three studies looked at CBT-based groups: in refugee populations [23, 24] and in a Brazilian favela [22]. Other studies evaluated interventions focused on expressive writing [34], mindfulness [35], online game-based mindfulness [39], trauma-focused CBT (delivered by lay counsellors) [38], a targeted group for sexual and gender minority adolescents (SGMA) [25], and a psychoeducational programme about posttraumatic growth [41].

**CBT-based groups.** Seven studies evaluated CBT-based groups. Four studies investigated Teaching Recovery Techniques (TRT), a five-week programme based on Trauma-Focused Cognitive Behavioural Therapy (TF-CBT) designed for children in low resource settings, making it suitable for delivery by non-clinicians. The program was developed by the Children and War Foundation [47]. El-Khani et al. [24] evaluated TRT in a RCT with Syrian refugee children aged 9–12 and their families in Lebanon, comparing it with a waitlist control group and a TRT + parenting (TRT+P) condition, which was rated as strong for quality. Data were collected pre-intervention (T1) and post-intervention at 2 weeks (T2) and 12 weeks (T3). All three conditions experienced a significant reduction in symptoms as measured on the Intrusion sub-scale on the CRIES-13, with the greatest reduction in the TRT+P condition, showing the lowest levels at T3. Both the TRT and TRT+P condition also experienced reductions in depression. Furthermore, all three conditions experienced a significant reduction in anxiety; however, the waitlist group's scores increased by T3. The authors suggest that a parenting skills component can enhance the positive effects of TRT. A pilot study by the same authors evaluating TRT in the same population in a Turkish setting was also included [31]. Similar findings were observed; a significant reduction was reported by participants in the Intrusion sub-scale scores on the CRIES-13 at two weeks post-intervention, although other measures did not reach significance owing to low power (n = 16).

Sarkadi et al. [45] evaluated TRT in an adolescent refugee population in Sweden and found significant reductions in PTSD (CRIES-8) and depression (MADRS-S) symptoms two weeks after the intervention. Qualitative data indicated that the programme had utility through provision of social support, normalisation, valuable tools, and manageability. Barron [22] found similar results in their RCT evaluating the impact of TRT for youth with PTSD in Brazilian favelas, showing a significant reduction in PTSD and depression symptoms, and a small effect size for posttraumatic growth compared with the control group. TRT was the most extensively studied intervention and has the strongest evidence base for reducing PTSD symptoms of all papers included in this review [22, 24, 31, 45].

Bryant et al. [23] conducted a RCT evaluating a similar group-based intervention called Early Adolescent Skills for Emotions (EASE) for Syrian adolescents. The intervention spanned seven group sessions focusing on arousal reduction, behavioural activation, and problem management. Data were collected pre-intervention (T1) and post-intervention at 9 weeks (T2) and 3 months (T3). Significant reductions in the PSC-internalising scale were found compared to the control group, and this effect was sustained at T3. EASE was also evaluated by Akhtar et al. [21] in a pilot RCT, although the study was low powered and did not detect any significant changes.

Elswick et al. [32] evaluated the impact of a group after school club called the Trauma Healing Club, based on Cognitive Behavioural Interventions for Trauma in Schools (CBITS). CBITS is an evidence-based intervention deliverable by a clinician, whilst the Trauma Healing Club is suitable for non-clinicians. Significant improvement in PTSD symptoms were found, with mean scores falling from above the threshold for clinical intervention to below. This study did not report on withdrawals, the reliability and validity of measures, or potential confounders.

Li et al. [32] assessed the Power up Children's Psychological Immunity (PCPI) program, a modified group version of TF-CBT. This study also did not report on reliability and validity of measures, or potential confounders. Participants reported significantly reduced PTSD, depression, and anxiety following the intervention, but by the 3 month follow up, levels were similar to pre-intervention levels.

Overall, CBT-based groups had the strongest evidence base of any intervention type, with three RCTs rated strong or moderate quality providing promising evidence. The programmes TRT and EASE are most likely to deliver impact in reducing PTSD symptoms.

**Other CBT-based approaches.** Four studies explored CBT-based approaches in three distinct contexts: one-to-one interventions [30, 38], an online platform [36] and an art-based curriculum [40].

Murray [38] conducted an evaluation of TF-CBT for children affected by HIV/AIDS in Zambia. This was a modified lay counsellor-delivered version of the intervention, typically administered by clinicians. The sessions involved a combination of individual sessions with the child, caregiver(s) separately, and family sessions. Lay counsellors, lacking formal qualifications in counselling or mental health, typically provide psychological support through a community setting, often with a faith-based focus. A significant reduction in the severity of trauma symptoms and shame symptoms was found when comparing pre-intervention and two weeks post-intervention scores. The authors did not provide a detailed description of how TF-CBT was adapted for non-clinician delivery, but it included cultural adaptations and simplified terminology. Close weekly monitoring was provided by the trainer and a supervisor to ensure fidelity.

Jaycox et al. [36] assessed an online stress and trauma curriculum called Life Improvement for Teens (LIFT), which incorporated elements resembling TF-CBT and CBITS. The self-paced, internet-based curriculum was delivered online to high school students, who accessed the intervention during a weekly slot at the school site. LIFT comprised both a stress track and a trauma track, with students who reported potentially traumatic experiences being allocated

to the trauma track. This study found significant improvements in PTSD symptoms and emotional problems but observed no change in depression or anxiety. This study did not specify the validity of data collection tools, and it did not discuss reasons for participant withdrawal.

Dumornay et al. [30] evaluated the impact of a peer delivered trauma-informed CBT skills curriculum among young men aged 17–24 years involved in the justice system. The intervention was delivered by 'Paraprofessionals' such as youth workers and volunteers in community settings, often within participants' homes or on the street, to maximise engagement. This study was of weak quality, with a high likelihood of selection bias and unreliable or unverified data collection tools. Although improvements in distress related to employment and education over time were found, emotion regulation did not show significant improvement.

Sitzer and Stockwell [40] also evaluated an art programme which incorporated aspects of CBT. This is discussed in the subsequent Art and community-based approaches section.

**Mindfulness.** Three studies focussed specifically on evaluating a mindfulness intervention [28, 35, 39]. Ito et al. [35] assessed the effects of a short-term mindfulness-based group intervention delivered within a school setting for adolescents with trauma. Participants attended a single 60-minute group session and were subsequently encouraged by teachers to continue using mindfulness skills and cognitive defusion techniques. The study revealed significant improvements in mindful attention and awareness, along with reductions in depression, anxiety symptoms, and post-traumatic stress symptoms related to hyperarousal, recorded two weeks after the intervention. This study was rated moderate quality. Schuurmans et al. [39] compared the impact of three meditation-based online games among youth in a residential care setting. Participants engaged in twelve 15-minute sessions twice a week for 6 weeks. One game, called Muse, demonstrated improvements in post-traumatic symptoms, stress, and anxiety at the one-month follow-up, while results for the other games were inconsistent. Although the study had a moderate overall quality rating, it featured a very small sample size, with n = 15 at randomisation and n = 9 at the one month follow up. Davis and Aylward [28] evaluated the impact of a trauma-informed mindful yoga intervention, which was delivered remotely to high school students. However, the study did not find significant differences in depression, anxiety, or resilience following the intervention.

**Art and community-based approaches.** Five studies evaluated interventions based on art, culture, or community: two cohort design studies [27, 40] and three qualitative studies [42–44]. Sitzer and Stockwell [40] assessed a Wellness Programme delivered in schools, drawing on elements of art therapy along with aspects of CBT and Dialectical Behavioural Therapy (DBT). Results indicated significant increases in resilience and social and emotional functioning, particularly among male students. However, only unvalidated measures were used in this study. Barnett et al. [27] assessed the impact of residential culture camps on the wellbeing of Alaska Native youth aged 13 to 18 years. The results suggested a significant increase in positive affect, self-esteem, and a sense of belonging. However, not all measures in this study were validated.

Harden et al. [43] examined the effects of a twice weekly after school programme involving media and theatre activities as youth-led empowerment strategies, conducted on a university campus. The study identified positive themes related to empowerment, post-traumatic growth, and peace restoration among participants. Nevertheless, there were numerous methodological issues. The stated aim of the study was 'to contribute to a narrative that would reflect the [Truth N' Trauma] project's value', which is biased. There was no consideration of the relationship between researchers and participants, or ethical issues. These methodological weaknesses were common across all included qualitative studies.

Özden Bademci et al. [44] also evaluated a programme held on a university campus, featuring creative workshops and mentoring from volunteer university students for homeless boys

aged 14–17 years. Qualitative themes indicated that participants perceived the campus as a safe, engaging environment, developed trusting relationships with mentors, and increased their capacity for emotional regulation. However, similar methodological critiques applied, including a lack of consideration for the potential ethical issues arising from the mentoring relationships, and insufficient detail provided concerning the interview protocol. McMahon & Pederson [42] assessed a community-based juvenile justice diversion program and identified positive themes related to self-efficacy, empathy, connection, and conflict resolution. However, the research protocol, including how themes were reached, was not described.

**Other interventions.** The remaining interventions, which did not fit into thematic categories, will be summarised individually. Greenbaum & Javdani [34] evaluated a therapeutic writing intervention called Writing and Reflecting on Identity To Empower Ourselves as Narrators (WRITE ON), delivered to juvenile justice-involved youth. The intervention consisted of 90-minute sessions twice a week for six weeks. The study observed significant increases in positive mental health attributes and resilience throughout the programme and at a two week follow up.

Goldbach et al. [25] conducted a RCT to assess a 10-week intervention called Proud & Empowered (P&E), which targeted SGMA and was conducted during the school day. Participation in the P&E intervention reduced minority stress and improved depression and suicidality measures, although there was no significant change in PTSD symptoms.

Taku et al. [41] examined a psychoeducational intervention about post-traumatic growth (PTG) in three conditions: a Control Group, Group 1 (intervention focused on stress-related reactions and PTG), and Group 2 (focused on negative changes and PTSD only). In Study 1, Group 1 and the Control Group showed higher PTG than Group 2 after two weeks, suggesting that exposure to negative information about PTSD may supress PTG. In Study 2, there were no differences in PTG perceptions between the groups. Findings were inconclusive for this moderate quality study.

Eruyar and Vostanis [33] evaluated the impact of group Theraplay, an attachment-based intervention, with refugee children and their parents. After eight weekly sessions, a significant improvement was found in PTSD scores. However, only 59% of participants completed the programme, and reasons for withdrawals were not provided. Additionally, not all data collection tools were validated. This study was one of seven studies included in this review with interventions involving both children and caregivers, all of which suggested a positive impact [24, 31, 33, 38, 45].

Day et al. [29] assessed a trauma-informed intervention in a residential school, involving curriculum changes and access to relational interventions such as Theraplay to build attachment, self-esteem, and trust in others. Provision was made for two rooms to act as alternative environments for struggling students, providing access to problem solving, talk therapy, and use of sensorimotor activities. Significant differences were found in the pre and post test scores for post-traumatic symptoms. However, the study was weak in quality given the high chance of selection bias and a lack of validated measures. Furthermore, the intervention was broadly described without specific details.

Martin and Wood [37] evaluated a group drumming intervention delivered in a school and found significant improvements for boys in terms of higher mental wellbeing and lower post-traumatic stress symptoms, but not for girls. This study was of weak quality due to selection bias and a high unexplained dropout rate.

## Discussion

A systematic review methodology was used to investigate the evidence base for trauma-based interventions which can be used to support school-age youth affected by ACEs and which are

suitable for delivery by non-clinicians. This is the first systematic review to comprehensively appraise interventions in this domain specifically designed for non-clinician delivery. The most frequently studied CBT-based intervention was TRT [22, 24, 45]. Several studies evaluating TRT were excluded earlier in the review process because they were administered by a clinician. While TRT itself is suitable for non-clinician delivery, some excluded studies involved clinician delivery to enhance fidelity. CBT-based groups, particularly TRT, are suggested as the interventions in this domain with the strongest evidence base. EASE is another CBT-based group for which two studies found evidence of effectiveness, and which could be an effective intervention [21, 23]. CBITS and TF-CBT are two evidence-based group trauma recovery interventions [48]; although these interventions are beyond the scope of this review as they are clinician-delivered, four studies found evidence of the effectiveness of adapted versions of these widely recognised programmes delivered by non-clinicians [26, 32, 36, 38], which is promising. CBT groups are likely to be a good target for future research into interventions suitable for non-clinicians. EASE and TRT are two manualised interventions which may be suitable for this.

Two online interventions which delivered significant change included aspects of CBT [36, 39], and three further studies involved interventions which included aspects of CBT into their design, such as problem-solving [29], relaxation and breathing [32], developing a trauma narrative [34]. Although none of the included studies articulated a theory of change as to why the intervention might work, these aspects of CBT may underlie mechanisms for why interventions work. Interventions involving these activities are therefore worthy of further study. Whilst only inconsistent evidence was found for the effectiveness of art-based programmes, research indicates that art therapy is an effective modality for young people to engage with [49]. Furthermore, both TRT and EASE suggest that participants in the group can engage with activities through drawing their ideas rather than speaking or writing [50, 51]. Therefore, further research in this area is suggested, to establish if art-based interventions can be successfully adapted for delivery by non-clinicians.

Several of the included studies trialled interventions with refugee populations [21, 23, 24, 31–33, 45]. This area has a growing evidence base, considering that refugees are often supported in low-resource environments such as refugee camps, where accessing specialised support can be challenging even after achieving settled status. Notably, this review excluded studies specifically measuring war- or disaster-related trauma. As a result, the studies which have been included from this context are generalisable to a wider range of young people affected by adversity.

Six studies evaluated interventions which involved caregivers in some or all of the sessions [21, 24, 31, 33, 38, 45]. El-Khani et al. [24] observed that TRT achieved a greater positive impact on young people's mental health and PTSD symptomology when a parenting component was included. This finding suggests a potential avenue for future research, given the intergenerational nature of ACEs and the recognised role of trusted relationships with close adults as a potential protective factor for youth exposed to ACEs [52]. This may be of relevance to primary schools, which often have closer relationships with parents and caregivers and may be well positioned to trial interventions which include family members.

This review has made an important contribution to research and practice in identifying an emerging evidence base for trauma-based interventions for youth affected by ACEs which are suitable for delivery by a non-clinician. ACEs are associated with mental health problems in the short term [53] and a wide range of physical and mental health problems in the long term [2]. Schools have been identified as an essential part of the wider mental health system [7], yet information on interventions offered by schools is rarely collected and even less commonly evaluated [54]. Evidence-based interventions in schools have the potential to reach young

people affected by ACEs and offer impactful support early, which may in turn prevent unfavourable outcomes across the lifetime. Effective support, delivered early, in this area may mitigate against the long-term health and financial costs of ACEs [4].

## Implications for research

CBT-based groups and interventions involving caregivers should be prioritised for future research assessing non-clinician interventions. Assessing adaptations of existing clinician-delivered evidence-based interventions for supporting young people with ACEs (such as CBITS or TF-CBT) could be a good starting point for this. Further research into mindfulness or writing-based interventions could also be beneficial, or including these elements within a CBT group. Rather than relying solely on clinician-delivered designs to ensure fidelity of interventions intended for non-clinicians, more robust results can be derived from studies that implement non-clinician delivery, supplemented with measures such as observation or clinical supervision to maintain fidelity. Additionally, it is essential to incorporate longer-term follow up into study design. Most studies included in this review collected post-intervention data within a period of 2 weeks or less. Longer-term follow up assessments should be implemented in future research to establish the robustness of intervention evidence.

A lack of high-quality studies was observed, highlighting the need for further RCTs to assess the potential impact of interventions. Only nine out of the twenty-five studies identified were rated as strong or moderate quality. In situations where conducting RCTs may not be feasible, efforts should be directed towards ensuring the reliability and validity of data collection tools. Furthermore, researchers should diligently record and follow up on withdrawals from the study. This practice is both methodologically and ethically important, as traumatised youth may disengage from therapeutic interventions if they perceive them as harmful or unsupportive. Exploring this perspective can provide valuable insights into improving intervention strategies.

## Implications for educational settings

Based on available evidence reviewed here, the approaches which are most likely to be effective at supporting school-age youth who have experienced ACEs are CBT-based groups or mindfulness approaches. Additionally, interventions involving caregivers may also have value in this context. This aspect might be particularly pertinent for primary schools, which serve pupils aged under 11 years in the United Kingdom and are often able to liaise more closely and regularly with parents and caregivers. Educational settings should take these findings into account in selecting interventions for pupils. Schools are often unable to access specialist clinician-delivered support for students [55]. It is also common for school to lack resources or access to training for supporting staff to implement targeted interventions for students at risk of trauma [56]; this review provides actionable suggestions for interventions which are most likely to achieve impact in low-resource environments when delivered by non-clinical staff.

## Review limitations

The overall quality of this systematic review was assessed using AMSTAR 2 [57]. The review meets or partially meets all checklist items on AMSTAR 2. Although the time frame for follow up was not stated in the inclusion criteria, it has been reported for all strong and moderate quality studies. The review did not examine study registries and did not seek to consult experts in the field, and sources of funding for the studies included in the review have not been reported. Given the high level of anticipated and actual heterogeneity of studies in this area, a meta-analysis was not planned or conducted, which restricts the scope of the conclusions that can be drawn.

The search terms used in our review to retrieve studies on trauma-based interventions were limited, with participant ACE exposure identified as part of inclusion criteria for study selection but terms for ACEs not included in our search strategy. Trauma is a commonly used term in ACE-related research and interventions, and our strategy enabled a manageable number of studies to be retrieved and identified a broad range of studies meeting our criteria. However, it may have missed studies that used alternative terminology (e.g. those focused on specific adversity types) and further reviews using broader terminology would be beneficial in strengthening knowledge. Finally, the work only examined studies from the past ten years, only selected those published in English, and may have missed those published in non-academic literature. The secondary screen of the references of papers may have help to identify some of the articles missing due to the search criteria, but future work would be needed to identify interventions published in grey literature or published in languages other than English.

## Conclusions

Our review of twenty-five studies revealed emerging evidence for non-clinician delivered interventions for enhancing mental health outcomes in school-age youth with exposure to ACEs; specifically, evidence for the effectiveness of CBT-based group and interventions involving caregivers. There are several interventions which appear promising but which could benefit from further development and a more rigorous evaluation process. Further evaluation of the CBT-based groups, mindfulness-based approaches, and interventions involving caregivers, is recommended. As schools and educational institutions are increasingly expected to assume a greater role in supporting the mental health of youth exposed to ACEs, the development of this evidence base is of crucial importance.

## Supporting information

**S1 Table. Quality assessment tables.**
(DOCX)

**S1 File. Search strategy.**
(DOCX)

**S2 File. Data extraction headings.**
(DOCX)

**S3 File. PRISMA checklist.**
(DOCX)

## Author Contributions

**Conceptualization:** Mark Bellis, Karen Hughes, Sinead Brophy.

**Data curation:** Flo Avery, Natasha Kennedy, Michaela James, Hope Jones.

**Formal analysis:** Flo Avery, Natasha Kennedy, Michaela James, Hope Jones, Rebekah Amos.

**Funding acquisition:** Mark Bellis, Sinead Brophy.

**Investigation:** Flo Avery, Michaela James.

**Methodology:** Flo Avery, Michaela James, Hope Jones, Rebekah Amos, Karen Hughes, Sinead Brophy.

**Project administration:** Flo Avery.

**Supervision:** Mark Bellis, Karen Hughes, Sinead Brophy.

**Validation:** Natasha Kennedy, Rebekah Amos.

**Writing – original draft:** Flo Avery.

**Writing – review & editing:** Natasha Kennedy, Michaela James, Hope Jones, Rebekah Amos, Mark Bellis, Karen Hughes, Sinead Brophy.

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
