## [Decision Letter · Decision Letter 0]

19 Jan 2024

PONE-D-23-32516A systematic review of non-clinician trauma-based interventions for school-age children and young peoplePLOS ONE

Dear Dr. Brophy,

Thank you for submitting your manuscript to PLOS ONE. After careful consideration, we feel that it has merit but does not fully meet PLOS ONE’s publication criteria as it currently stands. Therefore, we invite you to submit a revised version of the manuscript that addresses the points raised during the review process.

Specifically, the reviewers request clarification and additional explanation of the methodology, the aim of the study, and the study selection process.

We look forward to receiving your revised manuscript.

Kind regards,

Jennifer Tucker, PhD

Staff Editor

PLOS ONE

“This work was supported by the National Centre for Population Health and Wellbeing Research. Funding was provided by Public Health Wales and the Economic and Social Research Council (ESRC) through their suport of a PhD studentship”

3. Thank you for uploading your study's underlying data set. Unfortunately, the repository you have noted in your Data Availability statement does not qualify as an acceptable data repository according to PLOS's standards.

5. Please include a copy of Tables 1-4 which you refer to in your text on pages 8 and 9.

Reviewers' comments:

Reviewer's Responses to Questions

**Comments to the Author**

1. Is the manuscript technically sound, and do the data support the conclusions?

Reviewer #1: Yes

Reviewer #2: Partly

2. Has the statistical analysis been performed appropriately and rigorously? 

Reviewer #1: N/A

Reviewer #2: No

3. Have the authors made all data underlying the findings in their manuscript fully available?

Reviewer #1: Yes

Reviewer #2: No

4. Is the manuscript presented in an intelligible fashion and written in standard English?

Reviewer #1: Yes

Reviewer #2: Yes

5. Review Comments to the Author

Reviewer #1: Check some spelling errors in the context, and correct. correct your referencing style as it is a bit confusing, see comments in the main document. I suggest that a table to unpack the methodology be used after the results. See suggestion in the document

Reviewer #2: The review question and aim is not direct.

study selection process can be difficult to replicate and some aspects are missing i.e appraisal.

Analysis method is missing and result presentation section is a description of some selected studies and not necessarily presented in terms of the study question.

This made the discussion section contain more "findings like" content and not necessarily discuss the findings.

The authors continuously mention that they included "weak studies" this leaves the reliability of the review in question.

6. PLOS authors have the option to publish the peer review history of their article (what does this mean?). If published, this will include your full peer review and any attached files.

Reviewer #1: No

Reviewer #2: No

---

## [Author Response · Author response to Decision Letter 0]

28 Feb 2024

Please see the attached Excel file for changes

---

## [Decision Letter · Decision Letter 1]

27 Mar 2024

PONE-D-23-32516R1A systematic review of non-clinician trauma-based interventions for school-age youthPLOS ONE

Dear Dr. Avery,

Thank you for submitting your manuscript to PLOS ONE. After careful consideration, we feel that it has merit but does not fully meet PLOS ONE’s publication criteria as it currently stands. Therefore, we invite you to submit a revised version of the manuscript that addresses the points raised during the review process.

**Kindly pay particular attention to the revisions suggested in the Introduction and Discussion**==============================

Kindly submit your revised manuscript by May 11 2024 11:59PM. If you will need more time than this to complete your revisions, please reply to this message or contact the journal office at plosone@plos.org. Please include the following items when submitting your revised manuscript:A rebuttal letter that responds to each point raised by the academic editor and reviewer(s). You should upload this letter as a separate file labeled 'Response to Reviewers'.A marked-up copy of your manuscript that highlights changes made to the original version. You should upload this as a separate file labeled 'Revised Manuscript with Track Changes'.An unmarked version of your revised paper without tracked changes. You should upload this as a separate file labeled 'Manuscript'.

We look forward to receiving your revised manuscript.

Kind regards,

Gerard Hutchinson, MD

Academic Editor

PLOS ONE

Reviewers' comments:

Reviewer's Responses to Questions

**Comments to the Author**

1. If the authors have adequately addressed your comments raised in a previous round of review and you feel that this manuscript is now acceptable for publication, you may indicate that here to bypass the “Comments to the Author” section, enter your conflict of interest statement in the “Confidential to Editor” section, and submit your "Accept" recommendation.

Reviewer #2: (No Response)

2. Is the manuscript technically sound, and do the data support the conclusions?

Reviewer #2: Partly

3. Has the statistical analysis been performed appropriately and rigorously? 

Reviewer #2: N/A

4. Have the authors made all data underlying the findings in their manuscript fully available?

Reviewer #2: Yes

5. Is the manuscript presented in an intelligible fashion and written in standard English?

Reviewer #2: Yes

6. Review Comments to the Author

Reviewer #2: (No Response)

7. PLOS authors have the option to publish the peer review history of their article (what does this mean?). If published, this will include your full peer review and any attached files.

Reviewer #2: No

---

## [Author Response · Author response to Decision Letter 1]

2 Apr 2024

Please see response to reviewers document which has been submitted for this revision.

---

## [Decision Letter · Decision Letter 2]

3 May 2024

PONE-D-23-32516R2A systematic review of non-clinician trauma-based interventions for school-age youthPLOS ONE

Dear Dr. Avery,

Thank you for submitting your manuscript to PLOS ONE. After careful consideration, we feel that it has merit but does not fully meet PLOS ONE’s publication criteria as it currently stands. Therefore, we invite you to submit a revised version of the manuscript that addresses the points raised during the review process.

**ACADEMIC EDITOR: Kindly address the comments made by the two reviewers, who have questioned the Introduction and Discussion and the rigour of the methodology alongside other concerns. ** 

We look forward to receiving your revised manuscript.

Kind regards,

Gerard Hutchinson, MD

Academic Editor

PLOS ONE

Reviewers' comments:

Reviewer's Responses to Questions

**Comments to the Author**

1. If the authors have adequately addressed your comments raised in a previous round of review and you feel that this manuscript is now acceptable for publication, you may indicate that here to bypass the “Comments to the Author” section, enter your conflict of interest statement in the “Confidential to Editor” section, and submit your "Accept" recommendation.

Reviewer #3: (No Response)

Reviewer #4: (No Response)

2. Is the manuscript technically sound, and do the data support the conclusions?

Reviewer #3: Yes

Reviewer #4: Partly

3. Has the statistical analysis been performed appropriately and rigorously? 

Reviewer #3: N/A

Reviewer #4: N/A

4. Have the authors made all data underlying the findings in their manuscript fully available?

Reviewer #3: Yes

Reviewer #4: Yes

5. Is the manuscript presented in an intelligible fashion and written in standard English?

Reviewer #3: Yes

Reviewer #4: Yes

6. Review Comments to the Author

Reviewer #3: This is an interesting paper which identifies gaps in the current literature regarding trauma-informed interventions in non-clinical settings. However, there are a few minor suggestions for improvement before publication.

My main concern is the balance of information presented in the main article vs the appendix. I think the authors would benefit from comparing their manuscript with other published systematic reviews in this journal. For example, it is unusual to see tables breaking down the scores for quality assessment for each item in those checklists in the main paper - this takes up a lot of room which would be better served with more information about your search strategy. Instead, tables 2-4 should be in your appendix, with the overall ratings added to Table 1.

Whilst you have completed a PRISMA diagram, this currently isn't referenced anywhere in the text. Figure 1 should therefore be signposted at the beginning of your results.

- it doesn't make sense to include the information about data extraction in the 'review design' section (lines 98-100), especially as it is detailed in the 'study selection' section below - suggest removing.

- language of publication was restricted to which languages? This needs to be specified

lines 163-165 - you do not need to repeat that this paper was included or need the detail of it being uploaded to Covidence - this is too much detail for a paper. This can be removed - stating "resulting in the identification of one additional paper" is sufficient.

- the new text on line 192 should be in the methods, not the results. You need to have a section entitled 'data synthesis' in your methods section. This should also explain your approach to data synthesis e.g. did you do a narrative synthesis?

- overall, I felt like the overall importance of implications of this review are understated - both in the introduction and discussion. You point out several limitations and gaps in existing literature, but the discussion would benefits from including more specific examples of novel interventions or trials which could be conducted. Think about answering the question "so what?"

Reviewer #4: The authors present a review of non-clinician delivered interventions addressing trauma symptoms for school-age youth who have experienced adverse childhood experiences. This is an interesting topic and an area where a review of evidence would be of benefit. Their narrative synthesis of 25 articles provides limited evidence of intervention efficacy due to a lack of high quality research however CBT-based group interventions are suggested to have the most promising evidence of efficacy of the interventions reviewed, for reducing PTSD symptoms in youth exposed to adverse childhood events.

Authors should identify synthesis methods in the abstract.

The sentence beginning on line 43 is confusing “While not everyone who has ACEs necessarily experiences a traumatic event [4], an ACE score of 1 or more is associated with the development of trauma symptoms [5], and this relationship is even stronger where four or more ACEs have occurred [6].” I think by definition if you experience an ACE you experience a traumatic event. The previous sentence already outlines that 4+ ACEs is more strongly associated with trauma symptoms. I would suggest amending this sentence for clarity or deleting.

The authors go on in the second paragraph to discuss the importance of increasing ACE awareness. It is unclear if they are referring to ACE-related trauma symptoms or ACE incidence. Further, the suggestion of trauma-informed services being recommended is not linked to a reason why, for what have they been recommended? I suggest editing this section for clarity.

The authors refer to the target population as “youth with trauma” in line 59, I would suggest a more appropriate way to describe the population here would be ‘youth who have experienced potentially traumatic events’ or youth experiencing trauma-related symptoms – depending on the populations represented in the reviews being referred to. Throughout the review this is an important distinction to clarify – are you referring to universal interventions for any youth who has been exposed to potentially traumatic events or those experiencing trauma-related symptoms?

The review would benefit from more clarity around the research question. Are the authors interested in universal prevention of trauma symptoms in youth with any exposure to ACEs, or targeted interventions for youth experiencing trauma symptoms after exposure to ACEs?

The research question “what evidence exists for the efficacy of non-clinician delivered interventions for supporting trauma recovery or improvements in mental health in school-age youth (4-18 years) who have experienced ACEs?” suggests the authors’ target population are youth experiencing trauma symptoms (who were exposed to ACEs) and the inclusion criteria state indicate they looked for articles describing interventions targeting ‘recovery’ however they have not extracted data to indicate symptom severity. This is an important factor when reporting efficacy in reducing symptoms across different studies and interventions. Without giving the reader, the participants’ baseline trauma symptom severity or level of ACE exposure, it is difficult to compare the interventions. In the characteristics of studies table, some studies describe the measure for trauma/PTSD as “at least one ACE” – over 60% of the population have experienced at least one ACE. Comparing efficacy of interventions in populations with one ACE against participants with over 4 ACEs is very different, the authors should provide information on trauma symptom severity and take baseline severity into account when making suggestions on which interventions may be effective for which population groups. Alternatively, the authors could divide into two research questions – one addressing universal prevention interventions for youth exposed to ACEs and another for youth with trauma-related symptoms.

I have concerns that the search strategy was not aligned well with the research question and was potentially not comprehensive enough to provide a true overview of the evidence in the field. The authors state the target population is youth who have experienced ACEs, yet they have not included ACE-related search terms. I am concerned they may have missed relevant articles by not including these terms. If it is the authors intention to include articles with any ACE exposure but not required to be experiencing trauma-related symptoms, I suggest they re-run the searches with inclusion of ACE-related terms and terms for each ACE category (e.g. abuse, violence, neglect, maltreatment, assault, divorce, incarceration etc) to include studies that have assessed the efficacy of interventions for youth exposed to any ACEs. The trauma search terms are very limited, and studies may have been missed that use terms such as ‘distress’ or ‘posttraumatic’ or ‘mental health’.

The data extracted should include sample size and information on how long and often it was implemented e.g. length of intervention, number of sessions, length of sessions, frequency and who delivered – this is one of the key characteristics of the review – the non-clinician delivery, therefore the reader is interested in what type of non-clinician delivered the intervention.

The purpose in the characteristics of studies should have the comparator added for RCTs, it would also be helpful for the reader if table 1 included sample size and intervention setting, severity of symptoms/no. of ACEs and mode of delivery.

The paragraph starting on line 208 mixes mode of delivery (e.g. online) with intervention setting (e.g. school) – these should be described separately. Does residential setting mean in the participants home? If so, this would be a more informative description.

Line 235 introduces a paragraph describing the interventions assessed by the nine studies with moderate or strong quality evidence but only 8 are then listed.

Note RCT is an acronym, so the correct phrase is a RCT rather than an RCT.

Line 256 – “it is suggested..” do the study authors suggest this or the review authors – suggest clarifying

In the paragraph reporting results from Elswick et al. the authors do not provide any information of the efficacy of the intervention.

The overview of the current evidence and gaps in the field is incomplete. Several publications have reviewed school-based interventions for trauma symptoms. Reference to these could be useful to highlight the current article’s contribution to the literature e.g. Stratford et al 2020, Berger et al 2019, Chafouleas et al 2019.

The discussion is restating much of the results and should be revised to avoid re-presenting results but providing interpretation and comparison to existing literature and highlighting what the review add to the field. While the authors attempt to state whether the research question was answered, this is structured as though the question was to assess the quality of evidence followed by presentation of results. The discussion should be re-written so as to provide the reader with clarity on how the reviews finding answer the question and how this relates to existing evidence.

7. PLOS authors have the option to publish the peer review history of their article (what does this mean?). If published, this will include your full peer review and any attached files.

Reviewer #3: **Yes: **Dr Rebecca Appleton

Reviewer #4: No

---

## [Author Response · Author response to Decision Letter 2]

28 May 2024

Thank you for your comments. Please see attached file for full response

---

## [Editor Report · Decision Letter 3]

13 Jun 2024

A systematic review of non-clinician trauma-based interventions for school-age youth

PONE-D-23-32516R3

Dear Dr. Avery,

We’re pleased to inform you that your manuscript has been judged scientifically suitable for publication and will be formally accepted for publication once it meets all outstanding technical requirements.

Kind regards,

Gerard Hutchinson, MD

Academic Editor

PLOS ONE
---

## [Editor Report · Acceptance letter]

24 Jul 2024

PONE-D-23-32516R3 

PLOS ONE

Dear Dr. Avery, 

I'm pleased to inform you that your manuscript has been deemed suitable for publication in PLOS ONE. Congratulations! Your manuscript is now being handed over to our production team.

Kind regards, 

on behalf of

Dr. Gerard Hutchinson 

Academic Editor

PLOS ONE